# Arbitrary Generative Video Interpolation

**Guozhen Zhang**[1,†,*]  **Haiguang Wang**[1,†]  **Chunyu Wang**[2]    **Yuan Zhou**[2]
**Qinglin Lu**[2]    **Limin Wang**[1,3,‡]
[1]State Key Laboratory for Novel Software Technology, Nanjing University
[2]Tencent Hunyuan    [3]Shanghai AI Laboratory
zgzaacm@gmail.com  lmwang@nju.edu.cn

## Abstract

Generative Video Frame Interpolation (VFI), which synthesizes intermediate frames from a given pair of start and end frames, plays a pivotal role in video creation. However, existing generative VFI methods are constrained to producing a fixed number of intermediate frames, which significantly limits the flexibility in adjusting the frame rate or duration of videos during the creation process. In this work, we present **ArbInterp**, a novel generative VFI framework that enables efficient interpolation at any timestamp and of any length. Specifically, to support interpolation at any timestamp, we propose the Timestamp-aware Rotary Position Embedding (TaRoPE), which modulates positions in temporal RoPE to align generated frames with target normalized timestamps. This design enables fine-grained control over frame timestamps, addressing the inflexibility of fixed-position paradigms in prior work. For any-length interpolation, we decompose long-sequence generation into segment-wise frame synthesis. We further design a novel appearance-motion decoupled conditioning strategy: it leverages prior segment endpoints to enforce appearance consistency and temporal semantics to maintain motion coherence, ensuring seamless spatiotemporal transitions across segments. Experimentally, we build comprehensive benchmarks for multi-scale frame interpolation (2× to 32×) to assess generalizability across arbitrary interpolation factors. Results show that ArbInterp outperforms prior methods across all scenarios with higher fidelity and more seamless spatiotemporal continuity. Video demos are provided on the website: https://mcg-nju.github.io/ArbInterp-Web/.

## 1 Introduction

With the advancements enabled by large-scale pretraining, text-to-video and image-to-video generation models (Kong et al., 2024; Wang et al., 2025; Yang et al., 2024; Chen et al., 2025; Zhou et al., 2024) have demonstrated remarkable prowess in synthesizing realistic videos for complex scenes, enabling a wide spectrum of downstream applications. Among these, generative Video Frame Interpolation (VFI)—the task of generating coherent intermediate frames from given start and end frames—remains a fundamental and widely used function in video creation. Recent work has explored adapting pretrained video generation models for generative VFI, such as leveraging image-to-video models to produce high-quality interpolations conditioned on start and end frames (Wang et al., 2024; Feng et al., 2024; Xing et al., 2024b; Zhang et al., 2025; Xing et al., 2024a).

However, all existing works strictly adhere to the fixed interpolation paradigm, as shown in Figure 1, where a predetermined number of intermediate frames is generated from given start and end frames. This paradigm inherently hinders practical flexibility, as it precludes users from dynamically adjusting frame counts or frame rates (FPS) to meet specific needs. Moreover, unlike other conditional tasks (e.g., video prediction), VFI imposes unique requirements: it not only demands temporal consistency between intermediate and input frames but also requires fine-grained semantic understanding of input frames to generate plausible in-between frames. The fixed-frame paradigm, however, fails to fully model continuous motion dynamics—this is because it only accommodates fixed-frame-rate inputs.

---

*Work is done during internship at Tencent Hunyuan.
†Guozhen Zhang and Haiguang Wang contributed equally to this work.
‡Corresponding author.

Figure 1: A comparison between the fixed interpolation paradigm (a) and our proposed ArbInterp (b). ArbInterp enables flexible control of the temporal positions of generated intermediate frames by specifying any timestamps between 0 and 1.

This limitation thereby restricts the model's capacity to reason about coherent motion fields and synthesize smooth spatiotemporal transitions.

To overcome the aforementioned bottleneck, we introduce **ArbInterp**, a novel generative frame interpolation paradigm that enables frame generation specified at any timestamps and of any length, as illustrated in Figure 1 (b). Specifically, by defining the start frame at timestamp $t = 0$ and the end frame at $t = 1$, ArbInterp aims to synthesize intermediate frames for any target timestamp $t$ within this interval. This design allows flexible control over the temporal position of generated frames: for example, $t = [0, 0.5, 1]$ generates a single middle frame for 2× interpolation, while $t = [0, 0.25, 0.5, 0.75, 1]$ achieves 4× interpolation. What's more, by sampling only a small set of timestamps each time rather than requiring full video sequences, our approach substantially reduces computational costs while enhancing the generative model's fine-grained temporal awareness.

The key challenge lies in effectively injecting timestamp information into the generative model. To achieve this, we propose Timestamp-aware Rotary Position Embedding (TaRoPE), built on the insight that in existing DiT-based (Peebles & Xie, 2023) video generative models, each frame determines its relative position within the video sequence *exclusively* through temporal RoPE (Zhao et al., 2025). *This means that adjusting the temporal RoPE directly alters how the current frame perceives its actual position in the sequence.* By adapting the temporal RoPE to target timestamps, our method allows the model to perceive custom positions without introducing additional parameters. In practice, this requires only minimal fine-tuning to transfer pre-trained generative models to the frame interpolation task, demonstrating remarkable efficiency and generality.

Theoretically, treating the interval between start and end frames as a continuous motion field enables our approach to generate an infinite number of interpolations. Moreover, by specifying timestamps incrementally, long-term frame generation can be decomposed into sequential segments. For example, a long video interpolation with timestamps $[0, 0.01, 0.02, \ldots, 0.99, 1]$ can be divided into multiple segments, such as $[0, 0.01, \ldots, 0.10, 1]$, $[0, 0.11, \ldots, 0.20, 1]$, and so on. However, the inherent stochasticity of generative models can cause discontinuities in motion and appearance across segments. To address this, we further introduce a motion-appearance decoupling conditioning strategy to enhance spatiotemporal continuity between segments. Specifically, we feed the last frame of the previous segment as a conditioning input to preserve appearance consistency and inject motion information extracted from the last $N$ frames into the DiT forward process to maintain motion coherence. This dual injection mechanism significantly enhances cross-segment consistency, enabling high-fidelity infinite frame interpolation with seamless transitions in both visual appearance and motion dynamics.

Experimentally, we validate the effectiveness of the ArbInterp framework on the recently open-sourced video generation model of Wan (Wang et al., 2025). We find that fine-tuning for only 20,000 steps using 8 GPUs (96GB) is sufficient to achieve our goals. To rigorously assess generalizability, we construct comprehensive benchmarks spanning multi-scale frame interpolation tasks (e.g., 2×, 8×, 16×, and 32× interpolation). Experimental results show that ArbInterp outperforms prior methods across all tested scenarios: it achieves higher fidelity and seamless spatiotemporal continuity in different interpolation ratios from 2x to 32x. These results underscore our framework's superiority in balancing flexibility and generative quality for real-world VFI.

**Contributions.** In summary, the contributions of this paper are as follows:

- We propose a novel generative frame interpolation paradigm, ArbInterp, which can control the generated frames by specifying any continuous timestamps. By integrating timestamp

controllability into generative modeling, our approach demonstrates a superior flexibility and ability to model continuous dynamics.

- To achieve long-term frame interpolation, we design a novel motion-appearance decoupling strategy to efficiently enhance the spatiotemporal continuity between adjacent segments.

- We meticulously constructed comprehensive benchmarks encompassing multi-scale frame interpolation. Experimental results demonstrate that ArbInterp significantly outperforms previous models in both flexibility and performance.

## 2 RELATED WORK

### 2.1 VIDEO FRAME INTERPOLATION

Current video frame interpolation methods are broadly categorized into deterministic and generative approaches. Deterministic methods (Huang et al., 2022; Zhang et al., 2023; Reda et al., 2022), dominated by flow-based frameworks, typically predict intermediate optical flows between input frames and synthesize intermediate frames via warping operations. In these works, timestamps are typically integrated via optical flow scale factors or fed into networks as feature. However, constrained by limited data scales and model capacities, these methods struggle to accurately model motions in complex scenes. In contrast, generative models (Zhang et al., 2025; Jain et al., 2024; Feng et al., 2024; Wang et al., 2024; Xing et al., 2024b;a)—particularly those leveraging pre-trained video generation models adapted for interpolation tasks—exhibit superior generalization in challenging scenarios. These generative strategies can be further classified into two paradigms: (1) leveraging image-to-video conditional models (e.g., SVD) to generate videos conditioned on start and end frames separately and then merge the results into one video; and (2) latent-space conditioning methods, which integrate input frame information by concatenating or replacing latent codes and fine-tune models to guide intermediate frame synthesis. Despite their advancements, both categories adhere to a fixed-interpolation paradigm, restricting output to a predefined number of frames with uniform temporal spacing. In contrast, our method, ArbInterp, overcomes these limitations by enabling the generation of frames at arbitrary continuous timestamps between input frames.

### 2.2 RoPE IN VIDEO GENERATION

Rotary Position Embedding (RoPE) (Su et al., 2024) has become the prevailing approach in contemporary DiT-based video generative models (Zhuo et al., 2024; Wang et al., 2025; Yang et al., 2024), thanks to its precisely defined relative position information and outstanding performance. In most scenarios, the temporal position of each frame is simply its position within the current clip. For long-term generation, some methods endow frames with their true indices throughout the entire video (Guo et al., 2025; Zhang & Agrawala, 2025; Wu et al., 2024; Li et al., 2025; 2026). A recent study, RIFLEx (Zhao et al., 2025), demonstrates effective training-free length extrapolation by adjusting the intrinsic frequency in RoPE, highlighting the significance of RoPE in the temporal sequencing of generated videos. In this work, we introduce Timestamp-aware RoPE (TaRoPE), which assigns each frame a continuous timestamp within the range of 0 to 1 as its temporal position, rather than its index. This innovative design empowers the model to capture motions between input frames with infinite fine-grained precision and generate any frame at any timestamp.

### 2.3 LONG-TERM VIDEO GENERATION

Current approaches for long-video generation can be categorized into two distinct types. The first type involves decomposing a long video generation into the generation of multiple consecutive short segments (Henschel et al., 2024; Kim et al., 2024; Yin et al., 2024; Weng et al., 2024; Zhang & Agrawala, 2025; Villegas et al., 2022). The mainstream of such methods employs an autoregressive manner akin to frame-by-frame or segment-by-segment generation, with the key challenge lying in ensuring overall coherence and quality. For example, the recent work FramePack (Zhang & Agrawala, 2025) balances efficiency and performance by compressing the tokens of historical frames by manual strategy. The second type first generates anchor frames and then generates videos segmentally. Nuwa-XL (Yin et al., 2023), for instance, adopts a divide-and-conquer approach to achieve coarse-to-fine video generation. Notably, our proposed ArbInterp can seamlessly accommodate both of these two

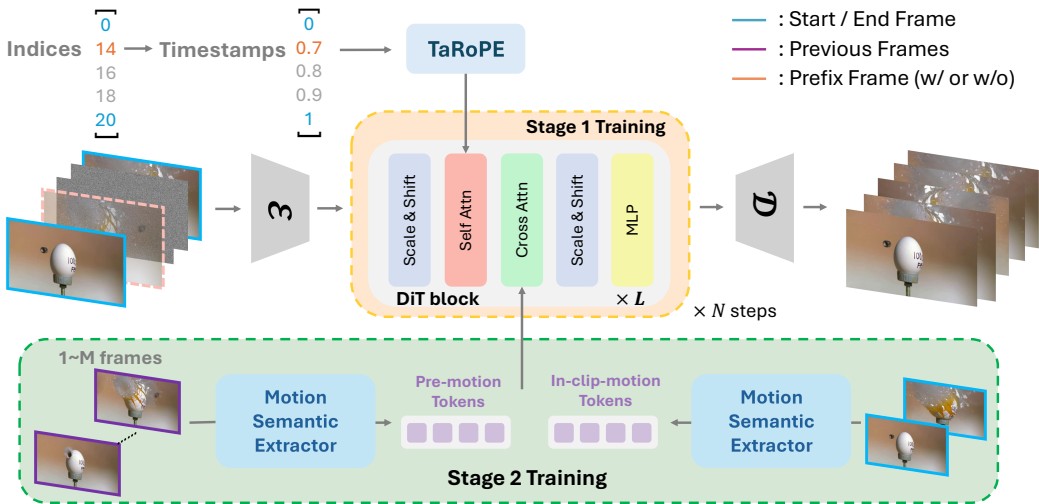

Figure 2: **Overall architecture of ArbInterp.** Our framework enables arbitrary-length interpolation with continuous timestamps using *Timestep-aware Rotary Position Embedding (TaROPE)*. Additionally, we introduce an *appearance-motion decoupling conditioning strategy* to enhance the performance of long-term interpolation. This strategy ensures appearance consistency via prefix frame guidance and enforces motion continuity through motion tokens.

designs, thus enabling effective and stable long-term video frame interpolation. Meanwhile, we further introduce a novel appearance-motion decoupled strategy to explicitly disentangle content and motion information, which effectively improves the spatiotemporal coherence between consecutive segments in long-term interpolation scenarios.

## 3 METHOD

### 3.1 ARBINTERP

Given the first frame $x_0$ and the last frame $x_1$, the goal of video frame interpolation is to generate intermediate frames. Previous works on generative frame interpolation (Wang et al., 2024; Feng et al., 2024; Xing et al., 2024b) can only produce a fixed number of frames, lacking flexibility and the ability to model continuous motion between frames at a fine-grained level. To address these limitations, we propose a novel framework, ArbInterp, which can generate intermediate frames at arbitrary temporal positions specified by continuous timestamps. Formally, given a timestamp list $T = [0, t_1, \ldots, t_n, 1]$ and input frames $x_0$ and $x_1$, ArbInterp generates corresponding intermediate frames as follows:

$$[x_{t_1}, \ldots, x_{t_n}] = \mathrm{ArbInterp}\,(x_0, x_1, T)\,, \tag{1}$$

where $t_i$ can be any value between 0 and 1. To leverage the powerful generative capabilities of pre-trained models, we build ArbInterp upon the recent open-source video generation model Wan (Wang et al., 2025). As shown in Figure 2, ArbInterp introduces two novel designs. First, we enable the denoising network to generate frames corresponding to specific timestamps by introducing Timestamp-aware Rotary Position Embedding (TaROPE), achieving fine-grained temporal modeling capabilities, as detailed in Section 3.3. Subsequently, to enhance the quality of long-term video interpolation, we design an appearance-motion decoupling conditioning strategy to strengthen the spatio-temporal coherence across different segments, as explained in Section 3.4. Based on these two designs, ArbInterp supports generating frames of arbitrary length at any continuous timestamps.

### 3.2 TRANSFERRING VIDEO GENERATION MODELS TO FRAME INTERPOLATION

For a video $x$, Wan (Wang et al., 2025) first performs spatio-temporal compression via a tokenizer (Kingma & Welling, 2022) to obtain video latents $z$. During training, for any sampled timestep $n \in [0, 1]$, Wan adds Gaussian noise $\epsilon^n \sim \mathcal{N}(0, I)$ to $z$ to produce noisy latents $z^n$, and trains a

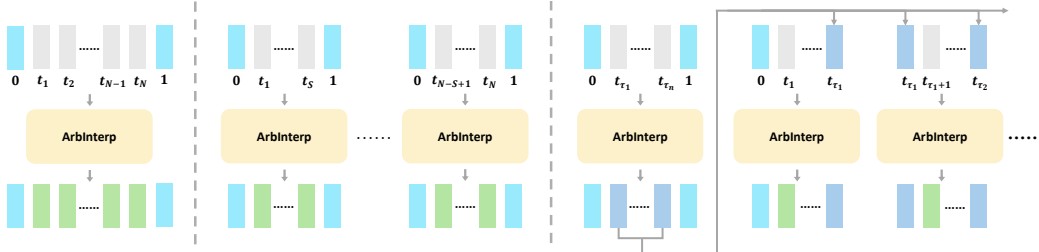

(a) Direct Interp.    (b) Segment-by-Segment Interp.    (c) Hierarchical Interp.

Figure 3: **Different interpolation strategies in ArbInterp.** ArbInterp supports multiple interpolation strategies: **(a) Direct Interpolation** for short-range interpolation, and two for long-term scenarios: **(b) Segment-by-Segment Interpolation** and **(c) Hierarchical Interpolation**.

denoising network using the flow matching framework (Lipman et al., 2023; Liu et al., 2023):

$$\mathcal{L} = \|\boldsymbol{v}_n - u_\theta(\boldsymbol{z}^n, n, \boldsymbol{y})\|^2, \quad \text{where} \quad \boldsymbol{v}_n = \boldsymbol{\epsilon}^n - \boldsymbol{z}, \tag{2}$$

where $\boldsymbol{y}$ denotes additional conditional information such as text, and $u_\theta$ represents the denoising network. The denoising network is composed of $L$ diffusion transformer blocks, each encompassing Adaptive Layer Normalization (AdaLN) (Peebles & Xie, 2023), self-attention, cross-attention, and a multi-layer perceptron (MLP). Self-attention facilitates information exchange within the video, while cross-attention incorporates auxiliary conditional information. To adapt Wan to the frame interpolation task, we adopt the token replace method in Open-Sora (Zheng et al., 2024), substituting the noisy latents of the first and last frames with the ground-truth latents to guide the prediction of intermediate frames. Additionally, to simplify the correspondence between each latent and its timestamp, we perform only spatial compression during tokenization. Since our method does not require predicting all frames simultaneously, the training cost remains computationally feasible.

### 3.3 TIMESTAMP-AWARE RoPE

#### 3.3.1 DEFINITION

**Vanilla Temporal RoPE.** To distinguish the spatio-temporal positions of each token, Wan applies 3D RoPE (Su et al., 2024) to each token in $\boldsymbol{z}$. Specifically, Wan partitions the channels of each token into three equal parts and applies RoPE for each dimension respectively. Since our work primarily focuses on temporal RoPE, we omit spatial RoPE in the following discussion. For the $k$-th latent $\boldsymbol{z}_k$ in the given video latent $\boldsymbol{z}$, the original temporal RoPE rotates it by $\theta_k$, as shown in Equation 3:

$$\tilde{\boldsymbol{z}}_k = \text{RoPE}(\boldsymbol{z}_k, k) = \boldsymbol{z}_k e^{i\theta_k}, \quad \theta_k = k\theta_{\text{base}}. \tag{3}$$

By applying temporal RoPE to both query ($Q$) and key ($K$) in self-attention, the attention score between any two latents at indices $k$ and $j$ is influenced by their relative positions:

$$A_{k,j} = \text{Re}[\langle \tilde{\boldsymbol{z}}_k, \tilde{\boldsymbol{z}}_j \rangle] = \text{Re}[\langle \boldsymbol{z}_k, \boldsymbol{z}_j \rangle e^{i(k-j)\theta_{\text{base}}}]. \tag{4}$$

Notably, temporal RoPE is the sole component in the denoising model that enables each frame's latent to perceive its temporal position, as all other operations are position-agnostic. This implies that altering the temporal RoPE assigned to each frame can modify its temporal position in the final generated sequence.

**Introduction of Timestamp.** In the default temporal RoPE, each latent's temporal position is represented by an absolute index, meaning the range of $|k - j|$ is fixed and always integer-valued when training with a fixed sequence length. For frame interpolation tasks, this leads the model to over-rely on latents at specific fixed positions. For example, when trained on 16 frames, the model tends to depend on the latents at positions 0 and 15, as these frames provide the actual conditional information. This rigid dependency restricts the model's ability to generalize to sequences of varying lengths. To address this issue, we propose *Timestamp-aware RoPE (TaRoPE)*, which specifies the temporal position of frames using continuous timestamps. Instead of using the latent's position in the input sequence, we adopt its real relative position between the first and last frames as the

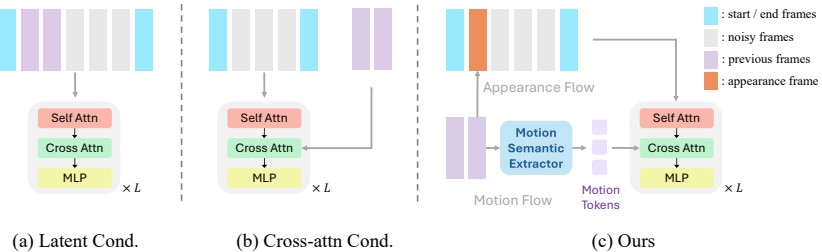

(a) Latent Cond.      (b) Cross-attn Cond.      (c) Ours

Figure 4: **Comparison of different conditioning strategies**: (a) direct latent conditioning, (b) cross-attention conditioning, and (c) our proposed appearance-motion decoupling conditioning strategy.

timestamp for temporal RoPE. For a sampled video with $N$ frames, the timestamp $t_k$ of the $k$-th frame is computed as:

$$t_k = \frac{k-1}{N-1}, \quad \text{s.t. } 1 \le k \le N. \tag{5}$$

As illustrated in Figure 2, the timestamp of the first frame is fixed at 0, and that of the last frame at 1. This approach normalizes frame interpolation for videos of arbitrary lengths into a continuous interval from 0 to 1. Consequently, the network learns to utilize information from latents at timestamps 0 and 1 to model the continuous motion field in a length-invariant manner.

### 3.3.2 TRAINING AND INFERENCE

To fully train the model's sensitivity to continuous timestamps, we adopt a segment-wise training strategy. During each sampling step, the model only predicts a segment of the complete video sequence. In practice, because TaRoPE treats videos of arbitrary length as continuous segments within $[0, 1]$, we can fully train on timestamps of varying granularities during training. For example, for a 200-frame video, we set the first and last frames as timestamps 0 and 1. When predicting frames 100 and 101, the input timestamp list becomes $[0, 1/2, 101/200, 1]$. As described in the appendix, our training involves predicting 1 to 19 intermediate frames, with a maximum interval of 2 seconds between the first and last frames. Given that the training set includes videos with frame rates ranging from 30fps to 120fps, the training timestamps naturally cover values from $1/2$ to $1/240$. Through this approach, the denoising network can effectively learn to generate corresponding frames from continuous timestamps. Concurrently, training costs are significantly reduced as there is no need to process entire video sequences.

In inference, timestamps are always normalized to $[0, 1]$, regardless of test length. For example, predicting $[0, 0.2, 0.4, 0.6, 0.8, 1]$ can be decomposed into $[0, 0.2, 0.6, 1]$ then $[0.6, 0.8, 1]$, ensuring test sequences never exceed the training maximum. Building on the design described above, ArbInterp supports multiple adaptive inference strategies to accommodate varying sequence lengths, as shown in Figure 3. For short sequences, we employ **direct interpolation**, generating the entire interpolated sequence in a single forward pass. For longer sequences with high computational demands, we introduce two approaches: **segment-by-segment interpolation**, where target timestamps are partitioned into non-overlapping segments processed sequentially; and **hierarchical interpolation**, which first predicts sparse anchor frames at coarse temporal intervals then refines the sequence by interpolating between these anchors. Hierarchical interpolation can better orchestrate global motion trajectories. However, segment-by-segment interpolation offers stronger real-time responsiveness in latency-sensitive scenarios (e.g., gaming). Both strategies reduce the computational complexity in self-attention from $O(N^2)$ to $O\left(\frac{N^2}{M}\right)$ when dividing a sequence of length $N$ into $M$ segments.

### 3.4 APPEARANCE-MOTION DECOUPLING CONDITIONING

Despite the effectiveness of segment-wise inference in ArbInterp for handling long-term interpolation, the stochastic nature of generative models introduces spatio-temporal inconsistencies between adjacent segments due to the indeterminate motion patterns between input frames. To address this challenge, specifically, our objective is to ensure spatio-temporal coherence between the current segment $s_i$ and its preceding segment $s_{i-1}$ by leveraging information from $s_{i-1}$. A straightforward solution, as shown in Figure 4(a), is to concatenate the latent of $s_{i-1}$ directly into the input. Al-

Table 1: **Quantitative comparison with the state-of-the-art methods on MultiInterpBench.** The **boldfaced** and underlined colors indicate the best and second best performing methods, respectively. ↑ indicates higher is better, ↓ indicates lower is better (applied to individual metrics in the header).

| Method | Interp. Rate | FID (↓) | FVD (↓) | LPIPS (↓) | CLIP$_{img}$ (↑) | VBench Metrics (↑ for all) | | | | | | |
| --- | --- | --- | --- | --- | --- | --- | --- | --- | --- | --- | --- | --- |
| | | | | | | Subject Consist. | Background Consist. | Temporal Flick. | Motion Smooth. | Aesthetic Quality | Imaging Quality | Overall Average |
| LDMVFI | | 85.8 | - | 0.297 | 0.863 | 0.9212 | 0.9198 | 0.9120 | 0.9403 | 0.4732 | 0.5901 | 0.7928 |
| TRF | | 108.5 | - | 0.435 | 0.879 | 0.8958 | 0.9060 | 0.8714 | 0.8744 | 0.4736 | 0.6224 | 0.7739 |
| GI | 2x | 90.8 | - | 0.496 | 0.893 | 0.9202 | 0.9156 | 0.8496 | 0.8527 | 0.4735 | 0.6252 | 0.7728 |
| DynamiCrafter | | 83.6 | - | 0.249 | 0.877 | 0.9228 | 0.9219 | 0.9189 | 0.9357 | 0.4829 | 0.6154 | 0.7996 |
| ArbInterp-SVD | | 59.1 | - | 0.152 | 0.902 | 0.9395 | 0.9473 | 0.9284 | 0.9475 | 0.4925 | 0.6314 | 0.8144 |
| ArbInterp | | 44.9 | - | 0.076 | 0.913 | 0.9590 | 0.9719 | 0.9353 | 0.9644 | 0.5027 | 0.6382 | 0.8286 |
| LDMVFI | | 62.2 | - | 0.232 | 0.881 | 0.9198 | 0.9371 | 0.9269 | 0.9609 | 0.4725 | 0.6036 | 0.8035 |
| TRF | | 59.0 | - | 0.410 | 0.877 | 0.8974 | 0.9035 | 0.9080 | 0.9378 | 0.4551 | 0.5932 | 0.7825 |
| GI | 8x | 62.5 | - | 0.505 | 0.887 | 0.9190 | 0.9146 | 0.9117 | 0.9264 | 0.4584 | 0.6118 | 0.7903 |
| DynamiCrafter | | 51.8 | - | 0.282 | 0.876 | 0.9246 | 0.9242 | 0.9387 | 0.9648 | 0.4701 | 0.6169 | 0.8066 |
| ArbInterp-SVD | | 40.6 | - | 0.183 | 0.893 | 0.9352 | 0.9497 | 0.9412 | 0.9695 | 0.4857 | 0.6342 | 0.8193 |
| ArbInterp | | 33.0 | - | 0.123 | 0.910 | 0.9529 | 0.9681 | 0.9485 | 0.9749 | 0.5036 | 0.6405 | 0.8314 |
| LDMVFI | | 49.6 | 397.2 | 0.281 | 0.897 | 0.9301 | 0.9242 | 0.9477 | 0.9790 | 0.4681 | 0.6057 | 0.8091 |
| TRF | | 47.3 | 530.4 | 0.407 | 0.871 | 0.8997 | 0.9255 | 0.9331 | 0.9609 | 0.4714 | 0.5973 | 0.7980 |
| GI | 16x | 53.4 | 534.2 | 0.516 | 0.879 | 0.9176 | 0.9279 | 0.9385 | 0.9562 | 0.4741 | 0.6116 | 0.8043 |
| DynamiCrafter | | 42.3 | 368.6 | 0.297 | 0.864 | 0.9266 | 0.9261 | 0.9515 | 0.9744 | 0.4703 | 0.6107 | 0.8099 |
| ArbInterp-SVD | | 35.3 | 279.8 | 0.205 | 0.898 | 0.9384 | 0.9446 | 0.9490 | 0.9801 | 0.4951 | 0.6359 | 0.8286 |
| ArbInterp | | 28.4 | 211.2 | 0.155 | 0.904 | 0.9486 | 0.9645 | 0.9572 | 0.9807 | 0.5077 | 0.6418 | 0.8334 |
| LDMVFI | | 52.6 | 753.6 | 0.358 | 0.817 | 0.9019 | 0.9320 | 0.9501 | 0.9695 | 0.4924 | 0.5997 | 0.8076 |
| TRF | | 50.8 | 1011.5 | 0.500 | 0.823 | 0.8751 | 0.9132 | 0.9274 | 0.9528 | 0.4538 | 0.5582 | 0.7801 |
| GI | 32x | 54.3 | 986.7 | 0.557 | 0.846 | 0.8963 | 0.9166 | 0.9291 | 0.9503 | 0.4566 | 0.5639 | 0.7855 |
| DynamiCrafter | | 46.4 | 732.7 | 0.374 | 0.825 | 0.8985 | 0.9334 | 0.9528 | 0.9750 | 0.4756 | 0.6009 | 0.8060 |
| ArbInterp-SVD | | 32.8 | 409.3 | 0.174 | 0.882 | 0.9247 | 0.9549 | 0.9562 | 0.9783 | 0.4973 | 0.6198 | 0.8219 |
| ArbInterp | | 26.5 | 319.9 | 0.145 | 0.906 | 0.9441 | 0.9628 | 0.9624 | 0.9817 | 0.5023 | 0.6411 | 0.8324 |

though effective, this method significantly increases computational overhead during both training and inference. Alternatively, another simple approach is to inject prior segment information through cross-attention (Figure 4(b)). While efficient, this approach yields weaker appearance consistency compared to direct latent concatenation (Zhang et al., 2024).

To balance efficiency and performance, we introduce the appearance-motion decoupled conditioning strategy, as shown in Figure 4(c). For appearance coherence, we use the last frame of $s_{i-1}$ as a prefix frame in the input to guide the generation of $s_i$. This minimal modification ensures visual continuity while incurring an acceptable computational cost. For motion coherence, inspired by the fact that modern generative models can control video motion using text prompts (e.g.,"rotating movement"), we extract semantic-level motion tokens from the last $N$ frames of $s_{i-1}$ using a *Motion Semantic Extractor* (MSE) to regulate the motion of the current segment. The MSE first employs a temporally enhanced CLIP model (Radford et al., 2021b) to extract spatio-temporal features aligned with the text semantic space. These features are then compressed into a fixed number of motion tokens via the Q-Former (Li et al., 2023a) to reduce redundancy. We enhance the temporal modeling of CLIP by adapting last $L$ layers of the original CLIP into spatio-temporal full attention augmented with temporal embeddings. Additionally, we also extract in-clip motion tokens from the start and end frames of $s_i$ using a shared MSE, enabling the model to better align generated motions with input constraints. Both types of motion tokens are injected into the network through the original cross-attention that interact with text prompts. When the cross-attention parameters are frozen, the MSE would learn to extract semantic information capable of controlling video motion, ensuring consistent motion dynamics across segments. More structure details and ablations of the MSE are provided in the supplementary materials.

## 4 EXPERIMENTS

### 4.1 IMPLEMENTATION DETAILS

We initialize ArbInterp based on the latest open-source video generation model Wan2.1-T2V-1.3B (Wang et al., 2025). Our training dataset comprises 50,000 videos meticulously curated from OpenVid (Nan et al., 2024). ArbInterp is trained on 8 GPUs (96GB) with a total batch size of 16. Inspired by He et al. (2025), we divide the training into three stages: In the first stage, we transfer the generation model to the frame interpolation task and finetune all parameters of DiT. The input includes start and end frames, a random number of intermediate frames to be predicted, and an optional prefix frame to ensure spatial consistency across different segments during testing. In the second stage, we freeze the denoising network and train the motion semantic extractor individually to

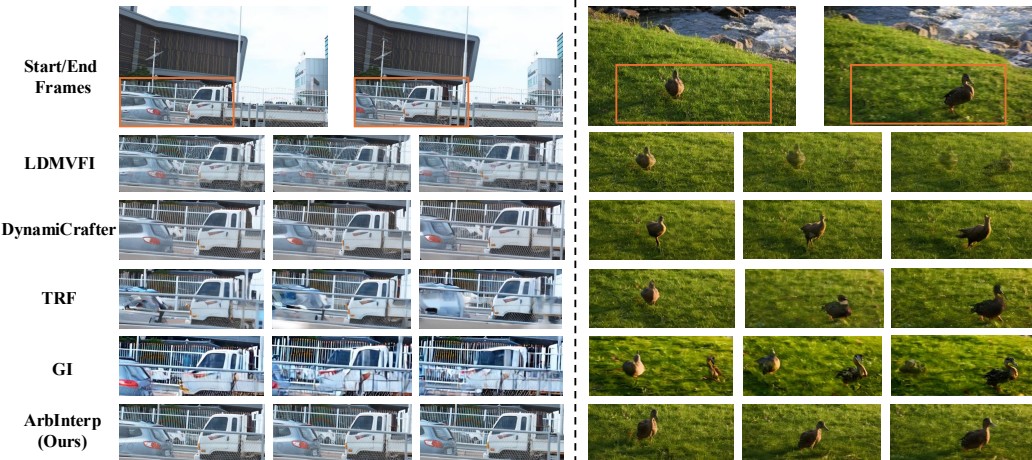

Figure 5: **Visual comparison.** The timestamps of the intermediate frames are 0.25, 0.5, and 0.75, respectively. ArbInterp demonstrates significant advantages in stability and consistency.

Table 2: **Ablation study on key designs of ArbInterp.**

| | Timestamp injection | Spatiotemporal continuity | Step-time (s) | FID | FVD | Subject Consist. | Background Consist. | Temporal Flick. | Motion Smooth. | Aesthetic Quality | Imaging Quality |
|---|---|---|---|---|---|---|---|---|---|---|---|
| 1 | None | N/A | 2.3 | 38.7 | 476.8 | 0.9193 | 0.9402 | 0.9415 | 0.9620 | 0.4833 | 0.6299 |
| 2 | Vanilla | N/A | 2.3 | 35.2 | 454.2 | 0.9210 | 0.9404 | 0.9439 | 0.9637 | 0.4852 | 0.6298 |
| 3 | TaRoPE | N/A | 2.3 | 33.7 | 401.6 | 0.9272 | 0.9519 | 0.9517 | 0.9707 | 0.4859 | 0.6296 |
| 4 | TaRoPE | Latent Cond. | 4.4 | 27.6 | 336.3 | 0.9390 | 0.9597 | 0.9601 | 0.9810 | 0.4922 | 0.6342 |
| 5 | TaRoPE | Cross-attn Cond. | 2.7 | 28.4 | 342.1 | 0.9381 | 0.9582 | 0.9584 | 0.9792 | 0.4935 | 0.6358 |
| 6 | TaRoPE | w/o Motion | 2.4 | 29.8 | 354.0 | 0.9376 | 0.9584 | 0.9549 | 0.9764 | 0.4937 | 0.6380 |
| 7 | TaRoPE | w/o Appearance | 2.3 | 31.7 | 382.7 | 0.9297 | 0.9533 | 0.9592 | 0.9803 | 0.4902 | 0.6331 |
| 8 | TaRoPE | Ours | 2.5 | **26.5** | **319.9** | **0.9441** | **0.9628** | **0.9624** | **0.9817** | **0.5023** | **0.6411** |

learn the ability to extract motion semantics. The motion semantic extractor takes up to 8 preceding frames as input. In the third stage, we train the entire model. The three stages are trained for 10k, 5k, and 5k steps, respectively. More implementation details are provided in the supplementary materials.

## 4.2 MULTIINTERPBENCH

To evaluate the flexibility and coherence in video interpolation at different interpolation ratios, we propose a new benchmark: *MultiInterpBench*. *MultiInterpBench* covers interpolation ratios of 2x, 8x, 16x, and 32x. For the first three ratios, ArbInterp employs direct interpolation in Figure 3(a), while for 32x, we adopt the strategy in Figure 3(c) combined with our proposed appearance-motion decoupling conditioning. The metrics include: (1) image-level metrics including LPIPS (Zhang et al., 2018), FID (Heusel et al., 2017), and CLIP similarity score ($CLIP_{img}$) (Radford et al., 2021a); (2) video-level metrics including FVD (Unterthiner et al., 2018; 2019) and VBench (Huang et al., 2024).

## 4.3 COMPARISON WITH THE STATE-OF-THE-ART METHODS

We compare ArbInterp with state-of-the-art generative frame interpolation methods: LDMVFI (Danier et al., 2024), DynamiCrafter (Xing et al., 2024b), TRF (Feng et al., 2024), and GI (Wang et al., 2024). Due to their inability to flexibly generate frames at arbitrary timestamps, we can only obtain their results through approximate strategies. Specifically, when exceeding their default frame counts, we use an iterative prediction strategy similar to Figure 3(c); when below, we use uniform sampling to select frames. Since most baselines are based on SVD-like architectures, we also trained a variant of ArbInterp based on SVD (Blattmann et al., 2023) (denoted as ArbInterp-SVD) for fair comparison. Unlike the DiT architecture, we interpolate the absolute positional encoding in SVD to enable generation at any timestamp.

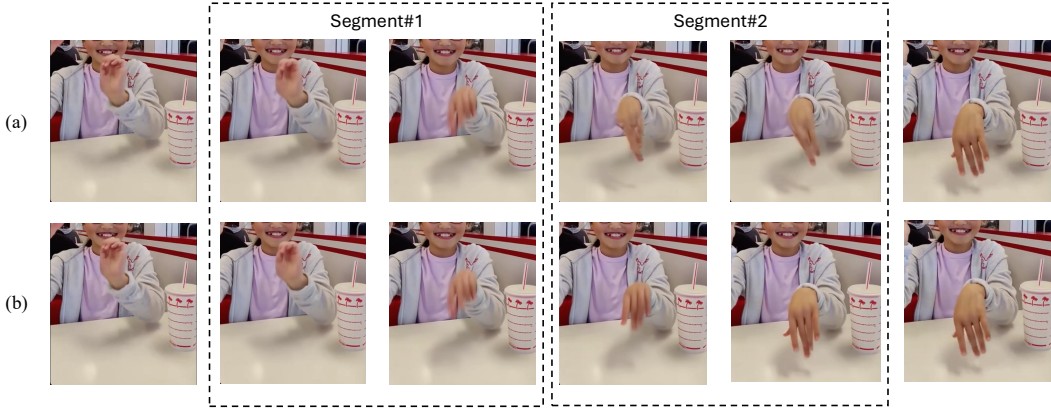

Figure 6: **Visual comparison of appearance-motion decoupling conditioning strategies.** (a) is produced by direct segment-by-segment prediction. (b) is the result with our proposed strategy.

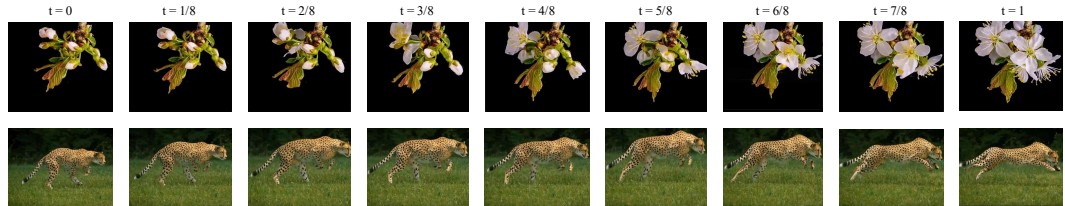

Figure 7: **Visualization of independently predicted intermediate frames at different timestamps.**

**Quantitative comparison.**    As shown in Table 1, benefiting from the flexibility and fine-grained timestamp awareness brought by TaRoPE, our model significantly outperforms previous methods across different interpolation ratios. Meanwhile, the performance superiority at 32x further demonstrates the advantage of ArbInterp in long-term interpolation.

**Qualitative comparison.**    As shown in Figure 5, compared to previous methods, our method can generate smooth and continuous intermediate frames solely by specifying timestamps, fully showcasing the flexibility and performance advantages of ArbInterp.

### 4.4    METHOD EXPLORATION

**Ablation study.**    As presented in Table 2, we conduct ablation studies on the core components of ArbInterp via performance analysis under 32x interpolation ratio. For timestamp incorporation, we compare two baselines: direct fine-tuning without timestamp awareness (None) and MLP-based injection (Vanilla). Meanwhile, we compare different strategies in Figure 4 and perform separate ablations on motion and appearance components, as shown in rows ID 4-8 of the table. Based on the ablation results, we can derive two key insights: (1) The TaRoPE approach outperforms the Vanilla baseline across all metrics, with a particularly pronounced improvement in motion smoothness. (2) Incorporating motion information yields substantial enhancements in temporal flicker and motion smoothness, while integrating appearance information significantly boosts subject and background consistency. In Figure 6, we further present a visual comparison, demonstrating the improvement in spatiotemporal coherence achieved by our proposed strategy. What's more, during inference, compared to latent concatenation, our strategy not only delivers better performance but also significantly improves computational efficiency by approximately 40%.

**Evaluation the capability of arbitrary-time interpolation.**    To demonstrate that ArbInterp equipped with the proposed TaRoPE is capable of directly generating intermediate frames corresponding to arbitrary continuous timestamps, we conduct an evaluation where intermediate frames are independently predicted at various continuous timestamps, with visual results presented in Fig. 7.

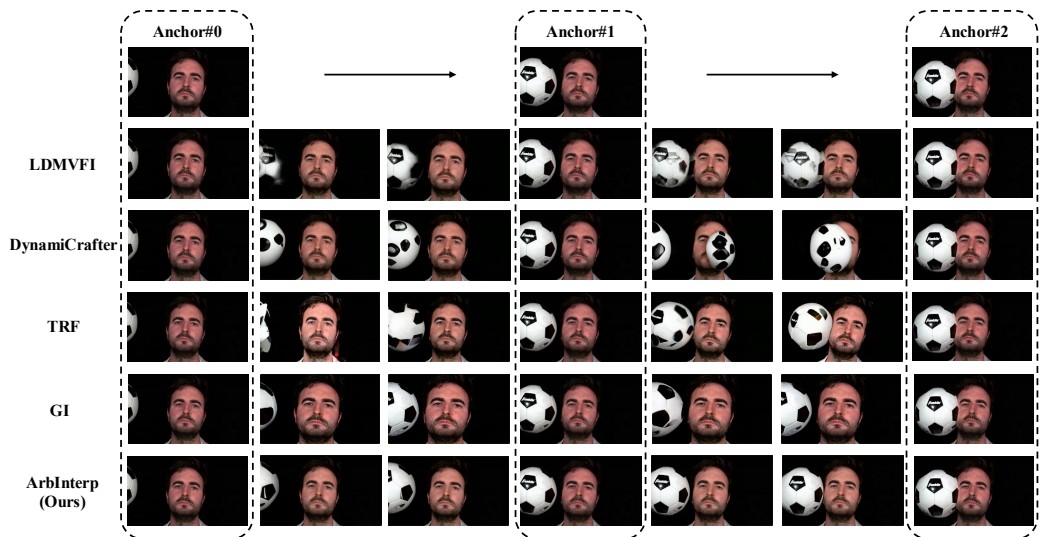

Figure 8: **Visual comparison in streaming frame interpolation scenarios.**

From the experimental results, it can be clearly observed that ArbInterp effectively captures the smooth, continuous motion evolution between the initial and final reference frames. By leveraging the temporal modeling capability introduced by TaRoPE, the model accurately infers the intermediate visual content and faithfully synthesizes realistic intermediate frames strictly aligned with the user-specified continuous timestamps. These results validate that our method achieves reliable arbitrary-frame interpolation without being restricted to discrete or equidistant intermediate steps.

## 4.5 EXTENSION TO STREAMING FRAME INTERPOLATION

The proposed design can naturally be extended to streaming frame interpolation scenarios. Specifically, for a sequence of consecutive input frame pairs $\{(x_0^i, x_1^i) \mid x_0^i = x_1^{i-1}\}$, we regard the frames from the preceding generated interpolation of $(x_0^{i-1}, x_1^{i-1})$ as the previous segment to aid the interpolation generation of $(x_0^i, x_1^i)$. By integrating our proposed appearance-motion decoupled conditioning strategy, ArbInterp efficiently enforce spatio-temporal coherence across sequential frame interpolation. **Notably, our method is the first effort in the generative video field to address the coherence challenge in streaming frame interpolation.**

For streaming frame interpolation, we generate 16 intermediate frames between every two consecutive anchors using direct interpolation. Meanwhile, the appearance-motion decoupling conditioning strategy is also employed to ensure spatiotemporal consistency across different anchors. For long-term streaming interpolation scenarios, as shown in Figure 8, ArbInterp exhibits superior motion coherence and appearance consistency by effectively leveraging previously generated frames to assist subsequent generation. More video comparisons are provided in the website.

## 5 CONCLUSION

In this work, we present ArbInterp, a novel generative frame interpolation paradigm that can generate an arbitrary number of intermediate frames by specifying continuous timestamps. To achieve this, we introduce *Timestamp-aware RoPE (TaRoPE)*, enabling the model to perceive the actual temporal position of each frame rather than relying on fixed input indices. Through this simple yet effective mechanism, ArbInterp theoretically supports infinite-length interpolation. Furthermore, we propose an *appearance-motion decoupling conditioning* strategy to enhance spatio-temporal consistency in long sequences. We design *MultiInterpBench* to rigorously evaluate ArbInterp. Experimental results demonstrate the superior flexibility of ArbInterp across various interpolation ratios and strong coherence of long-term video interpolation, offering new architectural possibilities for future generative frame interpolation research.

## ACKNOWLEDGMENTS

This work is supported by the Basic Research Program of Jiangsu (No. BK20250009) and the Collaborative Innovation Center of Novel Software Technology and Industrialization.

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

# Appendix

# A  MODEL DETAILS

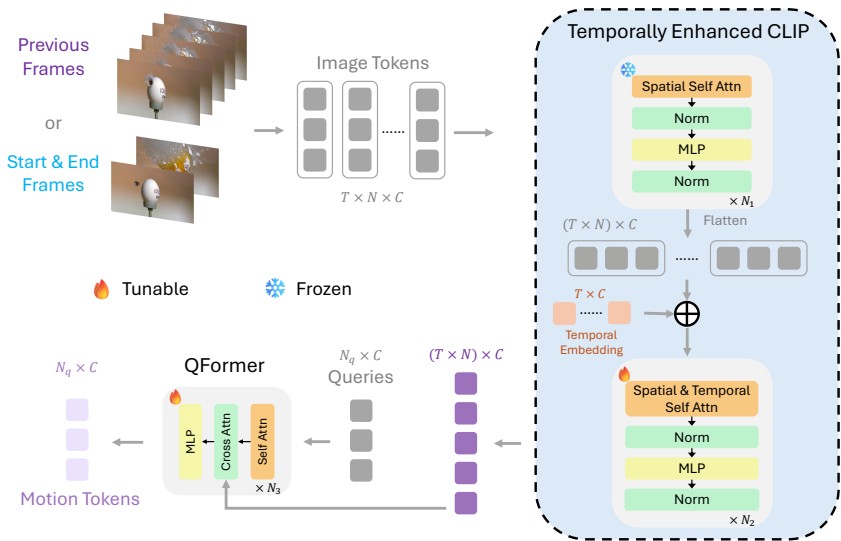

Figure 9: **Structure of the motion semantic extractor.** We first extract spatio-temporal features using the temporally enhanced CLIP, then compress these features into motion tokens via Q-Formers.

## A.1  DETAILS OF TRANSFERRING VIDEO GENERATION MODELS TO FRAME INTERPOLATION

In Section 3.2, we have already briefly introduced our strategy for transferring video generation models to frame interpolation. Here we provide more details.

Starting from an input video $\boldsymbol{x} = \{\boldsymbol{x}_1, \boldsymbol{x}_2, ..., \boldsymbol{x}_t\}$ with $t$ frames, we encode it with the video tokenizer (Kingma & Welling, 2022) provided by Wan (Wang et al., 2025), where each frame $\boldsymbol{x}_i$ is individually encoded without temporal compression to get frame-wise video latents $\boldsymbol{z} = \{\boldsymbol{z}_1, \boldsymbol{z}_2, ..., \boldsymbol{z}_t\}$.

During training stages, for any timestep $n \in [0, 1]$, we add a Gaussian noise to non-condition frames, namely, frames excluding starting and end frames (*i.e.* $\boldsymbol{z}_1$ and $\boldsymbol{z}_t$), as well as prefix frame (*i.e.* $\boldsymbol{z}_2$) if it exists. Formally, the noising add process can be define as

$$\boldsymbol{z}_i^n = n \times \boldsymbol{\epsilon}_i^n + (1 - n) \times \boldsymbol{z}_i, \quad 2 \le i \le t - 1, \tag{6}$$

where $\boldsymbol{\epsilon}_i^n \sim \mathcal{N}(0, I)$ is the Gaussian noise we add to each frame. Afterwards, we can obtain noisy video latents $\boldsymbol{z}^n = \{\boldsymbol{z}_1, \boldsymbol{z}_2/\boldsymbol{z}_2^n, \boldsymbol{z}_3^n, ..., \boldsymbol{z}_{t-1}^n, \boldsymbol{z}_t\}$, which is inputted into ArbInterp to predict velocity vector $u_\theta(\boldsymbol{z}^n, n, \boldsymbol{y})$. We calculate loss for **non-condition frames only** as

$$\mathcal{L} = \sum_{i=1}^{t} m_i \odot \|\boldsymbol{v}_n - u_\theta(\boldsymbol{z}^n, n, \boldsymbol{y})\|^2, \tag{7}$$

where $\boldsymbol{v}_n, \boldsymbol{y}, u_\theta$ is as explained in Equation 2, and $m_i \in \{0, 1\}$ indicates whether a frame is a non-condition frame.

## A.2  DETAILS OF MOTION SEMANTIC EXTRACTOR

The overall architecture of our *Motion Semantic Extractor* (MSE) is presented in Figure 9. Taking a video clip or start and end frames as input, MSE firstly encode each frame into image tokens independently. Afterward, the image tokens are fed into a temporally enhanced CLIP model, which is initialized from a pre-trained CLIP (Radford et al., 2021b). This enhanced model comprises $N_1 + N_2$ layers, each consisting of a self-attention module, an MLP module, and layer normalization. The initial $N_1$ layers focus on spatial feature extraction, where self-attention modules operate exclusively on tokens within each frame to capture spatial relationships. To be noted, the initial $N_1$ layers

are frozen to retain CLIP's ability to capture spatial feature. Following these $N_1$ layers, temporal embeddings are added to the features of all frames to provide explicit temporal positional information. The receptive field of self-attention in the subsequent $N_2$ layers is expanded to encompass tokens from all frames, enabling the modeling of global spatio-temporal information across the entire video segment. Finally, the temporally enhanced CLIP model outputs a spatio-temporal video embedding. Both the temporal embedding and the final $N_2$ layers are tunable to acquire new spatio-temporal modeling capabilities.

To further compress motion information, we utilize a trainable Q-Former (Li et al., 2023b), which comprises of initial queries, and $N_3$ layers, each containing a self-attention module, a cross-attention module, and an MLP module. The queries interact with the video embedding by cross-attention, which distills the essential motion information from the video embedding into a fixed number of output motion tokens, effectively compressing key spatio-temporal dynamics.

## B  MORE IMPLEMENTATION DETAILS

In Section 4.1 of the main paper, we provide an concise version of the implementation details. This supplementary section intends to provide more necessary technical details.

### B.1  EVALUATION DETAILS

To comprehensively evaluate the performance of ArbInterp under diverse frame interpolation settings, we design a new benchmarks: *MultiInterpBench*. *MultiInterpBench* evaluates the model's flexibility and performance in generating frames at different timestamps and varying numbers, covering interpolation ratios of 2x, 8x, 16x, and 32x. It consists of 552 frame pairs selected from widely used datasets such as DAVIS (Pont-Tuset et al., 2017), SNU-FILM (Choi et al., 2020), and XTEST (Sim et al., 2021).

To balance GPU memory usage and computational efficiency, we adopt the direct frame interpolation method shown in Figure 3(a) for videos with fewer than 21 total frames. For example, the timestamp list for 8x interpolation is $[0, 1/8, 2/8, 3/8, 4/8, 5/8, 6/8, 7/8, 1]$. For scenarios requiring more than 21 frames (e.g., 32x interpolation), we employ hierarchical design in from Figure 3(c) due to its superior stability. Specifically, we first predict the frame $x_{0.5}$ at $t = 0.5$, then generate two 15-frame segments from $x_0$ to $x_{0.5}$ and from $x_{0.5}$ to $x_1$, respectively. We utilize the appearance-motion decoupling conditioning strategy to enhance the spatiotemporal coherence between these two segments.

Following Wan (Wang et al., 2025), we employ UniPC (Zhao et al., 2023) to optimize our flow matching-based denoising scheduler during inferences. In all inferences, we denoise from Gaussian noise for 50 steps with timestep shift set to 5.0.

### B.2  TRAINING DETAILS.

We load the pretrained weight of Wan2.1 1.3B (Wang et al., 2025) to initialize ArbInterp. The model accepts two input components: (1) an interpolation sequence comprising 3 to 21 frames, including a fixed starting frame, a fixed ending frame, an optional prefix frame, and the remaining slots filled with noisy intermediate frames; (2) preceding motion conditioning frames consisting of up to 8 contextual frames extracted from adjacent video segments. During training, prefix frame is sampled at a rate of 50%. The time gap between the starting and end frame is limited to 2 seconds at most. All input frames are adaptively resized to one of five predefined resolutions ($480 \times 832$, $480 \times 640$, $480 \times 480$, $832 \times 480$, and $640 \times 480$) based on most matched aspect ratio. Specifically, we first calculate the original aspect ratio of the input image and then select the target resolution that maintains the closest matching aspect ratio from our predefined set. Our training dataset comprises 50,000 videos meticulously curated from OpenVid (Nan et al., 2024). ArbInterp is trained in 3 stages on 8 GPUs (96GB) with a total batch size of 16. For memory-efficient training, we employed the Zero Redundancy Optimizer stage 3 (ZeRO-3) strategy (Rajbhandari et al., 2020). The base optimizer is AdamW (Loshchilov & Hutter, 2019), configured with a learning rate of $1 \times 10^{-4}$, betas of $(0.9, 0.999)$, an epsilon of $1 \times 10^{-8}$, and a weight decay of 0.01. The learning rate performs a linear warmup (Goyal et al., 2017) over the first 1000 training steps.

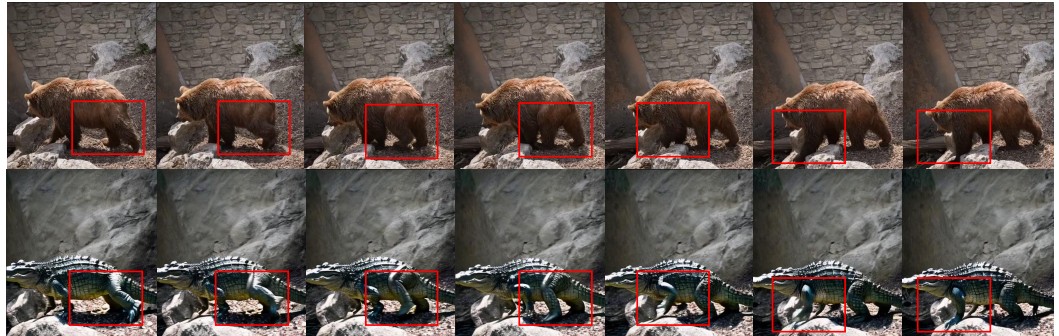

Figure 10: **Visualization of controlled motion transfer.**

**First stage.**   In the first stage, we only finetune the parameters of all DiT blocks and the output head for 10,000 optimization steps. The timestep shift of the noise scheduler is set to 1.0, ensuring uniform training across diverse input noise levels.

**Second stage.**   In the second stage, we freeze the parameters of DiT and train the motion semantic extractor, with trainable parameters including temporal transformer layers in CLIP, Q-Former layers, and MLPs for motion feature injection into DiT blocks. To enhance robustness, we implement randomized input dropout: 20% probability of discarding preceding motion conditioning frames, 20% probability of omitting boundary frames, 10% probability of dropping both inputs simultaneously, and 50% probability of retaining full inputs. This stage executes approximately 5,000 training steps. In this stage, the noise scheduler is configured with a timestep shift of 3.0 to increase the expected noise intensity, which enhances the motion feature injection's control over high-level semantic content. In the stage, with 50% likelihood, previous frames are chosen before the starting frame while intermediate frames are consecutive to and follow it; otherwise, previous frames and intermediate frames are consecutive sequences sampled after the starting frame, with intermediate frames directly succeeding previous frames. Such data sampling strategy aims at ensuring the model to be equally proficient at utilizing motion context either prior to or subsequent to the starting frame, both are required during inferences.

**Third stage.**   In the third stage, all parameters during the first and second stages are trained. This stage proceeds for 5,000 training steps. To be noted, we set the timestep shift of the noise scheduler back to 1.0 for balanced training over all noise intensities.

## C    ADDITIONAL RESULTS

### C.1    VISUAL COMPARISON OF TAROPE

We present visual comparisons, as shown in Figure 11, to further validate the effectiveness of TaRoPE. Methods without timestamp injection or those solely relying on MLP-injected AdaLN fail to precisely control the temporal position of generated intermediate frames. In contrast, our proposed TaRoPE accurately generates frames corresponding to specific timestamps.

### C.2    THE EFFECTIVENESS OF MOTION DECOUPLING

To rigorously verify the proper decoupling of appearance and motion in our framework, we trained a new dedicated model. This model reconstructs the original video by being conditioned on the first frame and leveraging our proposed Motion Semantic Extractor (MSE). During testing, we feed the video into the MSE to extract motion semantics, then edit the first frame (appearance) while retaining the extracted motion information, and observe whether the model can generate videos with consistent motion. As illustrated in the Figure 10, our decoupling mechanism successfully extracts motion information to a considerable extent, as evidenced by the generated videos that preserve the original motion despite modified appearance.

Table 3: **Quantitative comparison with the state-of-the-art methods on 256x interpolation.**

| Method | Interp. Rate | FID ($\downarrow$) | FVD ($\downarrow$) | LPIPS ($\downarrow$) | CLIP$_{img}$ ($\uparrow$) | VBench Metrics ($\uparrow$ for all) | | | | | | |
|--------|------|------|------|------|------|------|------|------|------|------|------|------|
| | | | | | | Subject Consist. | Background Consist. | Temporal Flick. | Motion Smooth. | Aesthetic Quality | Imaging Quality | Overall Average |
| LDMVFI | | 39.7 | 572.3 | 0.271 | 0.658 | 0.8568 | 0.8854 | 0.9026 | 0.9210 | 0.4678 | 0.5697 | 0.7672 |
| TRF | | 38.2 | 712.5 | 0.372 | 0.661 | 0.8313 | 0.8675 | 0.8810 | 0.9052 | 0.4311 | 0.5303 | 0.7411 |
| GI | 256x | 41.2 | 687.9 | 0.388 | 0.679 | 0.8515 | 0.8708 | 0.8827 | 0.9028 | 0.4336 | 0.5357 | 0.7462 |
| DynamiCrafter | | 34.5 | 553.1 | 0.297 | 0.663 | 0.8536 | 0.8867 | 0.9052 | 0.9263 | 0.4518 | 0.5709 | 0.7658 |
| ArbInterp-SVD | | 28.4 | 385.0 | 0.173 | 0.692 | 0.8835 | 0.9027 | 0.9097 | 0.9281 | 0.4733 | 0.5829 | 0.8144 |
| ArbInterp | | **21.5** | **242.3** | **0.118** | **0.728** | **0.8969** | **0.9147** | **0.9143** | **0.9326** | **0.4772** | **0.6090** | **0.7908** |

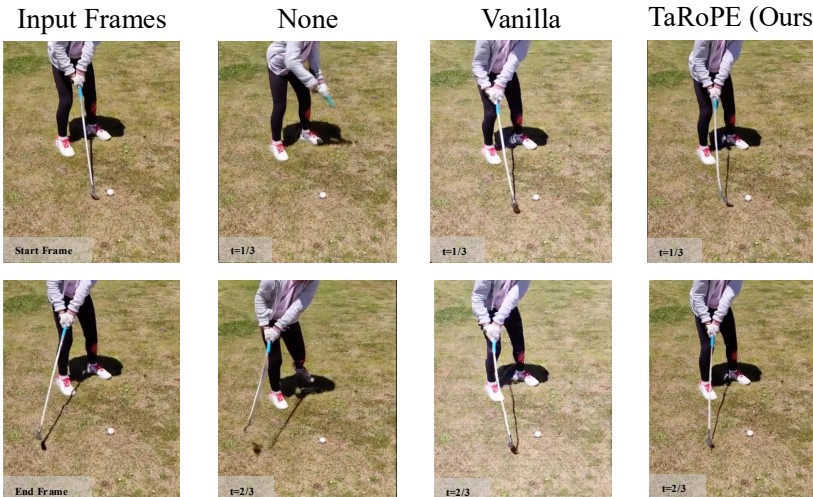

Figure 11: **Comparison of different timestamp injection methods.**

### C.3 INTERPOLATION RATIO BEYOND TRAINING

To further validate the ability to handle continuous timestamps, we provide 256× comparisons in Table 3. Since our original test set lacked sufficient frames, we re-collected 10 videos for this test. The result is obtained by averaging the results of each 32-frame sub-clip, as many metrics cannot process sequences of such length. Despite the granularity exceeding the training range, the sustained performance advantage further confirms the effectiveness of our method in handling continuous timestamps.

### C.4 ABLATION EXPERIMENTS ACROSS TRAINING STAGES

We also conducted ablation experiments on different training stages. As shown in Table 4, directly learning all functions (only stage3) fails to achieve the best performance. In contrast, the staged training approach, which first learns the basic frame interpolation function, then establishes spatio-temporal continuity between different segments, and finally performs joint fine-tuning, exhibits a higher performance ceiling.

## D LIMITATION AND FUTURE WORK

As a research-oriented work, the primary contributions of this work lie in successfully demonstrating that temporally fine-grained video frame interpolation can be achieved through introducing timestamp-aware RoPE into generative models, and in designing an appearance-motion decoupling conditioning strategy to facilitate long-term video frame interpolation. To maintain the simplicity of the overall design, we omitted text inputs and only used the first and last frames as inputs, which inevitably limits the model's controllability and semantic reasoning capabilities. Additionally, due to limited resources, we employed a relatively small generative model and a small-scale public dataset. Despite

Table 4: **Ablation study on different training stages.**

| Training Stages | | | Evaluation Metrics | |
|---|---|---|---|---|
| **Stage 1** | **Stage 2** | **Stage 3** | FVD$_{32x}$ ($\downarrow$) | VBench$_{32x}$ ($\uparrow$) |
| ✓ | | | 401.6 | 0.819 |
| ✓ | ✓ | | 359.2 | 0.8253 |
| | | ✓ | 357.2 | 0.8247 |
| ✓ | ✓ | ✓ | **319.9** | **0.8324** |

Table 5: Licenses and URLs for every benchmark, code, and pretrained models used in this paper.

| Assets | | License | URL |
|---|---|---|---|
| Benchmarks | OpenVid | CC-BY-4.0 | https://huggingface.co/datasets/nkp37/OpenVid-1M |
| | SNU-FILM | MIT license | https://github.com/myungsub/CAIN |
| | XTEST (-L) | for research and education only | https://github.com/JihyongOh/XVFI |
| Codes and Pretrained Models | Wan 2.1 | Apache-2.0 License | https://github.com/Wan-Video/Wan2.1 |
| | LDMVFI | MIT license | https://github.com/danier97/LDMVFI |
| | DynamiCrafter | Apache-2.0 License | https://github.com/Doubiiu/DynamiCrafter |
| | TRF | for research only | https://github.com/HavenFeng/time_reversal |
| | GI | Apache-2.0 License | https://github.com/jeanne-wang/svd_keyframe_interpolation |

outperforming state-of-the-art methods, the quality ceiling for complex scenarios remains constrained. In the future, we plan to integrate text guidance, scale both the dataset and model, and thereby advance the practical utility of our method for frame interpolation.

# E  BROADER IMPACT

This work proposes a novel generative frame interpolation paradigm, ArbInterp, which extends the capability boundary of current frame interpolation models. Similar to other generative models, although we use carefully screened data, the generated content still has a certain degree of uncontrollability. In the future, such uncontrollability can be constrained through further data cleaning and reinforcement learning. In this way, we can further balance the innovation of content with the potential impacts that fictionalization may incur.

# F  LICENSE OF DATASETS AND PRE-TRAINED MODELS

All the dataset we used in the paper are commonly used datasets for academic purpose. All the licenses of the used benchmark, codes, and pretrained models are listed in Table 5.

# G  USE OF LARGE LANGUAGE MODELS (LLMS)

This paper only uses LLMs for grammar correction.

