# OpenReview forum: "Arbitrary Generative Video Interpolation"
_ICLR.cc/2026/Conference — ICLR 2026 Poster_

### Official Review · Reviewer_E5JG · 2025-10-30

**Soundness:** 2
**Presentation:** 3
**Contribution:** 2
**Rating:** 6
**Confidence:** 2

**Summary:**

The paper proposes a method for continuous video interpolation that generates intermediate frames at any frame rate. Given a start and end frame, the method interpolates frames at arbitrary timestamps. To achieve this, the authors assign timestamps 0 and 1 to the start and end frames, respectively, normalize intermediate timestamps, and introduce a timestamp-aware RoPE. They further propose appearance–motion decoupled conditioning, enabling smooth and continuous segment-level interpolation. On VBench, the method achieves significantly superior video quality compared to prior approaches.

**Strengths:**

- **Novel problem setting:** This is the first work, to the best of my knowledge, to utilize generative models for continuous interpolation between two frames.
- **Clear and intuitive writing:** The paper is easy to follow and provides intuitive explanations. In particular, the concise extension from standard RoPE to timestamp-aware RoPE (TaRoPE) is elegant and highly effective. The method demonstrates strong performance, significantly outperforming existing approaches.

**Weaknesses:**

- While it is novel to apply generative modeling to VFI, methods that use continuous timestamps already exist in the non-generative video frame interpolation literature [1]. The paper does not compare against these methods. Including such comparisons would more clearly position this work relative to deterministic approaches.
- The authors combine timestamp-aware RoPE with a training dataset containing a wide timestamp range (1/2–1/240), which likely contributes significantly to performance. However, it is unclear whether the baselines were trained under the same settings. If baselines were trained only at fixed rates, it becomes difficult to attribute performance gains solely to TaRoPE rather than to broader timestamp coverage. It would be beneficial to train at least one baseline using the same diverse-rate data to clarify this.

[1] Super slomo: High quality estimation of multiple intermediate frames for video interpolation, Huaizu, et al., 2018

**Questions:**

- Beyond VFI, it would be valuable to verify whether appearance and motion have indeed been properly decoupled. If the decoupling is effective, could one borrow motion from other objects and use it as input for controlled motion transfer?
- Can the model also predict consecutive frames at non-equidistant timestamps? For example, at timestamps such as [0, 0.1, 0.4, 0.5, 0.9, 1]

---

> ### Author Response · Authors · 2025-11-24
>
> We sincerely appreciate your recognition and valuable comments. To thoroughly address your concerns, we provide the following detailed responses:
>
> ### Q1: Including comparisons with deterministic approaches
>
> **R1**: As presented in the table below, we report the performance of Super SloMo on our 32× test dataset. It is evident that ArbInterp achieves distinct performance advantages over deterministic methods for arbitrary frame interpolation, owing to its utilization of robust generative priors.
>
> | Setting            | FVD₃₂ₓ (↓) | VBench₃₂ₓ (↑) |
> | ------------------ | ---------- | ------------- |
> | Super SloMo | 896.5      | 0.7884        |
> | Ours        | 319.9      | 0.8324        |
>
>
>
> ### Q2: Train at least one baseline using the same diverse-rate data
>
> **R2**: To meet this request, we fine-tuned DynamiCrafter (denoted as DynamiCrafter-ft) under the identical training settings with the diverse-rate dataset, as shown in the table below. Although DynamiCrafter-ft does achieve a certain degree of performance improvement compared to its original version, ArbInterp still maintains notable performance advantages. The core reason lies in the absence of our temporally adjustable TaRoPE module in DynamiCrafter, which restricts its ability to adapt to diverse temporal rates effectively.
> | Setting            | FVD₃₂ₓ (↓) | VBench₃₂ₓ (↑) |
> | ------------------ | ---------- | ------------- |
> | DynamiCrafter | 732.7      | 0.8060        |
> | DynamiCrafter-ft | 635.2      | 0.8139        |
> | Ours        | 319.9      | 0.8324        |
>
> ### Q3: Verify whether appearance and motion have indeed been properly decoupled
>
> **R3**: To rigorously verify the proper decoupling of appearance and motion in our framework, we trained a new dedicated model. This model reconstructs the original video by being conditioned on the first frame and leveraging our proposed Motion Semantic Extractor (MSE). During testing, we feed the video into the MSE to extract motion semantics, then edit the first frame (appearance) while retaining the extracted motion information, and observe whether the model can generate videos with consistent motion. As illustrated in the newly added Figure 10 in the Supplementary Materials (L972-L984), our decoupling mechanism successfully extracts motion information to a considerable extent, as evidenced by the generated videos that preserve the original motion despite modified appearance.
>
> ### Q4: Can the model also predict consecutive frames at non-equidistant timestamps?
>
> **R4**: To verify this capability, we conducted targeted experiments by splitting the 8× prediction timestamp list. Specifically, we divided the equidistant sequence \[0, 1/8, 2/8, …, 1] into two non-equidistant timestamp sequences: \[0, 1/8, 3/8, 4/8, 6/8, 1] and \[0, 2/8, 5/8, 7/8, 1], and used them to predict consecutive frames (Sep. Predition). The performance comparison results are presented below. Although our model experiences a slight performance drop compared to its performance under the original equidistant timestamp design, it still achieves significant advantages over the baseline (without TaRoPE). This result fully demonstrates the feasibility of TaRoPE in handling non-equidistant timestamps.
>
> | Setting            | FID$_{8x}$ (↓) | VBench$_{8x}$ (↑) |
> | ------------------ | ---------- | ------------- |
> | No TaRoPE | 44.7      | 0.8092        |
> | Ours$_{\text{Sep. Predition}}$ | 35.5      | 0.8287        |
> | Ours$_{\text{Origin}}$       | 33.0      | 0.8314        |

---

> > ### Comment · Reviewer_E5JG · 2025-11-25
> >
> > Thank you for the detailed response. After checking the experimental results, I find that my initial concerns regarding the deterministic approach, baselines trained with the same dataset, motion decoupling, and non-equidistant timestamps have been well addressed. Therefore, I will maintain my score.

---

### Official Review · Reviewer_PUsK · 2025-10-30

**Soundness:** 3
**Presentation:** 3
**Contribution:** 2
**Rating:** 4
**Confidence:** 5

**Summary:**

This paper aims to enable frame interpolation of arbitrary timesteps, which was not possible with pre-trained video diffusion model-based methods. The paper first propose TaRoPE, which is a variation of RoPE which normalizes the timestamp between the first and last frames, and enable predictions at continuous timestamps. To cope with high computation demands, the authors propose to make predictions segment-wise, either linearly (segment-by-segment) or hierarchically. In this process, to ensure spatio-temporal consistencies, appearance-motion decoupling conditioning is propoosed. The last frame tokens of a previous segment are used to ensure appearance consistencies between segments, and using a spatio-temporally tuned CLIP and a QFormer, the motion of the previous segment is encoded and this information is injected via cross attention.

**Strengths:**

1. The paper is easy to follow, with clear problem definition.
2. The proposed framework seems to be well-designed, with strong results.

**Weaknesses:**

1. Novelty of TaRoPE: the proposed approach of normalizing the timestamps sounds reasonable and seems to be effective. However, I feel hesitant to agree on the novelty of this scheme. To begin with, normalization of timestamps is a commonly used approach in the field of video frame interpolation (VFI) [1-6]. It does differ in that most methods used them for scaling the optical flows, while this paper used it in positional encoding instead of flows, but the idea itself of normalizing intermediate timestamps is not something very new in the field of (traditional) VFI. Second, normalization / use of decimal position coordinates in PE is something already discussed in [9]. For these reasons, I find the novelty of TaRoPE to be quite limited.

2. Evaluations: 1) This paper is missing several important state-of-the-art methods for comparison [11, 12]. 2) The evaluation is mainly conducted on a benchmark built by themselves, and does not evaluate on a common benchmark existing in the field, such as DAVIS and Pexels [10, 11]. Proposing a new useful benchmark is welcomed, but in order to justify the efficacy of one’s methodology, it is also important to show evaluations on a commonly used benchmark to ensure fairness. Regarding evaluations, I do not think the comparison against LDMVFI to be fair.

## On the definition of the task
I would like to suggest the task of the paper to be defined as either "generative inbetweening" or "keyframe interpolation", rather than video frame interpolation (VFI). This is not merely an issue of this paper only, but is an issue of all recent papers accommodating pre-trained video diffusion models for video frame interpolation. Despite using the same term “video frame interpolation”, the task these recent methods address are quite different from traditional VFI [1-8]. This recent VFI methods with pre-trained diffusion models [10-12] assume a more extreme, challenging scenario than the task of traditional VFI. Traditional VFI has been studied in the past decade to solve a scenario where the input video has a fps of {15, 30, 60, 120} fps and aim to increase this to {30, 60, 120, 240, …, 1000} fps [1-8]. However, recent studies based on pre-trained diffusion models, including this work, assume a scenario where the input frame rate is much lower, as low as 1 fps (25 frame gap) [10-12]. One may wonder how much of a big difference this could make, the input frame rate being 1fps vs 30fps. But I argue that this fps difference in fact does make a huge difference, changing the nature of the task. The traditional video frame interpolation task considers interpolation of 15+fps (usually 30fps) input videos. The time gap between frames is extremely small, approximately 0.03 seconds. This is an extremely short time and therefore the diversity of possible intermediate motions is very limited. On the other hand, when it comes to the scenario of using 1 fps input videos, the number of possible motions increase drastically within a 1 second gap. Accordingly, traditional VFI task is considered a “restoration” task, recovering intermediate data points from a set of densely sampled data points. On the other hand, the new extreme-case frame interpolation with pre-trained diffusion models is considered more of a “generation” task, since the given input data points are far more sparse. The two scenarios pursue clearly different directions, so I think it is not fair to compare pre-trained diffusion-based methods with traditional VFI methods in extreme-low-fps scenario. For instance, LDMVFI, one of the baselines in the paper, is trained under different settings with much smaller-size dataset. Theoretically speaking, I think there is a possibility that the experimental results could have been different under the same amount of computational cost and matched train data settings. Therefore, I think there should be a clear distinction between the two tasks and I suggest the term “keyframe interpolation” to be more appropriate for these extreme-case scenarios. “Generative interpolation” does not seem to be a good alternative, since there has been several generative model-based studies also in traditional VFI literature [13, 14]. If the paper were to adhere to the task of VFI, I suggest that the paper to also compare with traditional VFI methods in their 30 fps input scenarios too, for thorough and fair evaluations. I think this suggestion could partially resolve some of the issues I mentioned above, e.g., novelty of timestamp normalization.

[1] Jiang, Huaizu, et al. "Super slomo: High quality estimation of multiple intermediate frames for video interpolation." CVPR 2018

[2] Xu, Xiangyu, et al. "Quadratic video interpolation." NeurIPS 2019

[3] Niklaus, Simon, and Feng Liu. "Softmax splatting for video frame interpolation." CVPR 2020

[4] Sim, H., Oh, J., and Kim, M. "Xvfi: extreme video frame interpolation." ICCV 2021

[5] Kong, Lingtong, et al. "Ifrnet: Intermediate feature refine network for efficient frame interpolation." CVPR 2022.

[6] Huang, Zhewei, et al. "Real-time intermediate flow estimation for video frame interpolation." ECCV 2022

[7] Xue, Tianfan, et al. "Video enhancement with task-oriented flow." IJCV 2019

[8] Choi, Myungsub, et al. "Channel attention is all you need for video frame interpolation." AAAI 2020.

[9] Chen, Shouyuan, et al. "Extending context window of large language models via positional interpolation." arXiv preprint arXiv:2306.15595 (2023)

[10] Wang, Xiaojuan, et al. "Generative inbetweening: Adapting image-to-video models for keyframe interpolation, ICLR 2025

[11] Yang, Serin, Taesung Kwon, and Jong Chul Ye. "Vibidsampler: Enhancing video interpolation using bidirectional diffusion sampler." ICLR 2025.

[12] Zhu, Tianyi, et al. "Generative inbetweening through frame-wise conditions-driven video generation." CVPR 2025.

[13] Danier, Duolikun, Fan Zhang, and David Bull. "Ldmvfi: Video frame interpolation with latent diffusion models." AAAI 2024.

[14] Lew, Jaihyun, et al. "Disentangled motion modeling for video frame interpolation." AAAI 2025

**Questions:**

- I wonder the extrapolation ability and extension to other tasks of the proposed method. Despite the model being tuned for interpolation task only, it would be very interesting if the proposed method shows its effectiveness beyond interpolation task.

---

> ### Author Response · Authors · 2025-11-24
> **Part 1**
>
> We sincerely appreciate your insightful comments and constructive feedback. To address your concerns comprehensively, we present the following detailed responses:
>
> ### Q1: On the definition of the task and novelty of timestamp normalization
>
> **R1**: After careful deliberation, we  concur with your perspective that **"keyframe interpolation"** is a more precise terminological choice. As you correctly pointed out, our core objective is indeed to leverage the powerful generative priors from pre-trained generative models to tackle complex scenarios involving large motions between input frames. In line with your valuable suggestion, we will revise the relevant descriptions and remove the comparison with LDMVFI in the final version of our manuscript.
>
> Building on this acknowledgment of your viewpoint, we would like to further elaborate on the **novelty and contributions of TaRoPE**:
>
>
>
> 1. TaRoPE is the first work to propose a complete conceptual framework—coupled with **rigorous architectural design and comprehensive experimental validation**—for enabling arbitrary-time frame interpolation in the **keyframe interpolation task**. While the idea of arbitrary-time interpolation has been touched upon in traditional video frame interpolation (VFI) methods, as you noted, these approaches are not readily adaptable to pre-trained generative models. In contrast, our method is concise, effective, and fully compatible with the architectural paradigms of most contemporary generative models.
>
> 2. Although the work you cited (\[9]) has made insightful explorations on positional encoding in LLMs, this does not inherently validate its applicability in the video generation domain. Our method successfully demonstrates that a similar design philosophy remains effective for keyframe interpolation tasks. This achievement lays a solid foundational design for future research on leveraging pre-trained models in keyframe interpolation.
>
> We sincerely hope that the above discussions will help us reach a mutual understanding on this matter.

---

> ### Author Response · Authors · 2025-11-24
> **Part 2**
>
> ### Q2: On fair evaluations
>
> **R2**: To address your concerns about evaluation fairness, we would like to emphasize three key points:
>
>
>
> 1. **No universally accepted common benchmark exists to date**: While the works you cited (\[10, 11, 12]) all utilize DAVIS and Pexels as data sources, their test set selection criteria and the number of samples vary significantly. Consequently, these datasets cannot be regarded as a unified, standardized benchmark for fair comparison.
>
> 2. As elaborated in Lines 887–889 of our manuscript, our test set not only includes DAVIS but also incorporates SNU-FILM and XTEST—two widely recognized benchmark datasets in the video interpolation community specifically designed to evaluate performance on complex motions. This ensures our evaluation framework possesses sufficient community credibility.
>
> 3. To further alleviate your concerns, we have conducted additional comparative experiments between our method and \[11] on the **only publicly available test set proposed in \[12]**. As shown in the table below, our method still achieves notable performance advantages, thanks to the inherent flexibility of arbitrary-time frame interpolation:
>
> | Method          | LPIPS  | FID  | VBench | FVMD  | FVD| LPIPS | FID  | VBench | FVMD  | FVD |
> |-----------------|-------------------------|-----------------------|------------------------|------------------------|-----------------------|-------------------------|-----------------------|------------------------|------------------------|-----------------------|
> |    Frame Gap    | 23      | 23    | 23     | 23     | 23    | 12      | 12    | 12     | 12     | 12    |
> | DynamiCrafter| 0.3886                  | 52.66                 | 0.8410                 | 13221.9                | 978.9                 | 0.3839                  | 37.49                 | 0.8458                 | 11810.7                | 652.5                 |
> | TRF          | 0.3687                  | 42.76                 | 0.8438                 | 10458.0                | 823.4                 | 0.3742                  | 39.01                 | 0.8478                 | 10076.6                | 818.4                 |
> | GI          | 0.2155                  | 31.39                 | 0.8606                 | 5682.6                 | 524.0                 | 0.2615                  | 32.37                 | 0.8651                 | 4721.0                 | 565.8                 |
> | Vibidsampler [11]          | 0.2973                  | 39.97                 | 0.8651                 | 9721.4                 | 730.5                 | 0.3437                  | 36.85                 | 0.8515                 | 8992.0                 | 724.7                 |
> | FCVG [12]            | 0.1832                  | 24.05                 | 0.8619                 | 5607.2                 | 437.9                 | 0.2378                  | 22.77                 | 0.8672                 | 4537.4                 | 465.6                 |
> | Ours            | 0.1659                  | 20.71                 | 0.8892                 | 4894.0                 | 385.3                 | 0.2115                  | 19.83                 | 0.8930                 | 4125.7                 | 413.9                 |
>
>
> ### Q3: On the capability of extension to other tasks
>
> Theoretically, our method can be extended to **any-frame conditional generation task that requires at least two frames as conditional inputs**. This is because a minimum of two frames is essential for the model to capture the **motion granularity of time intervals**.
>
> For video extrapolation tasks, TaRoPE is applicable to scenarios where at least the first two consecutive frames are available to generate subsequent frames. By adjusting the timestamps assigned to the extrapolated frames, our approach may enable precise control over the motion amplitude of the generated content.
>
> That being said, we argue that keyframe interpolation is the most suitable task for exploring and validating TaRoPE's core capabilities. Unlike other video generation tasks, it provides well-defined start and end keyframes that clearly delineate the temporal scope of the entire video sequence. We fully acknowledge the significant research potential and interest of these extensions. We will include a dedicated discussion on this topic in the *Further Work* section of our final manuscript, encouraging the research community to explore these promising directions in future studies.

---

> > ### Comment · Reviewer_PUsK · 2025-11-27
> >
> > I thank the authors for the rebuttal.
> > I am glad that the authors agreed with my suggestion on the task definition (Q1), and I am satisfied with the thorough evaluations conducted on a common benchmark with previous methods (Q2). I expect these discussions and experiments on extension (Q3) be included and have the main manuscript updated.
> >
> > Yet, some concerns still remain.
> >
> > 1) I am still not fully-convinced with TaRoPE's novelty, and think it is quite limited. (Q1) According to the authors' arguments, it sounds like the task difference is the only novelty, rather than technically. Applying something or validating something's effectiveness at another task could be a novelty if it has tailored components, or if it is an unexpected thing to perform well, but in the current form, it does not seem to be the case in both of them.
> >
> > 2) The authors claim that their proposed method can theoretically be applied to any-frame conditional generation task. (Q3) However, they did not show any experimental evidence on this. I agree that the proposed method has the potential to be applicable to other generation tasks, and expect it to display some possibilities.
> > Given the unresolved novelty issue remaining, I think this experiment could be helpful. If the proposed method could successfully be applied to other generation tasks, I think the uniqueness and importance of this paper could be elevated to a higher level.

---

> > > ### Author Response · Authors · 2025-11-27
> > >
> > > We sincerely appreciate your recognition of our previous discussions and experimental results. We provide clearer clarifications to elaborate on the uniqueness of our work and kindly request your further consideration:
> > >
> > > Compared to [9], ArbInterp indeed features tailored components—specifically, the capability for discontinuous segment generation. While [9] fully validates that positional interpolation can enable sufficient generation under long contexts, it lacks any discussions or experiments on segment-based generation.
> > >
> > > In ArbInterp, tailored to the inherent characteristics of video generation models, we decompose the generation of intermediate frames for long videos into multiple segments. Critically, TaRoPE enables unified control over the temporal relationships between these segments while preserving consistent motion granularity—this is a design uniquely optimized for video generation.
> > >
> > > Furthermore, we provide concrete visualizations to corroborate this capability. As shown in Figure 7, we independently predict each intermediate frame by adjusting TaRoPE, and the seamless motion continuity directly validates the effectiveness of our design. Additionally, we propose an appearance-motion decoupling strategy to further enhance continuity across distinct segments—an innovative attempt not explored in [9].
> > >
> > > We sincerely hope these discussions further clarify the contributions of our work. We fully agree that extending the proposed method to other generation tasks would significantly amplify its impact—and we will supplement results from a toy experiment within a short timeframe. We kindly ask for your patience as we finalize these additional results.

---

### Official Review · Reviewer_wVzm · 2025-10-31

**Soundness:** 2
**Presentation:** 2
**Contribution:** 2
**Rating:** 2
**Confidence:** 3

**Summary:**

This paper introduces a novel video frame interpolation pipeline that finetunes a pretrained video generation model. Conditioned on a start and end frame, the model synthesizes intermediate frames through a new rotary position encoding (RoPE) design and an appearance-and-motion decoupling mechanism. Experiments validate the efficacy of the proposed approach.

**Strengths:**

The paper is well written and easy to follow.
The paper presents strong experimental results that surpasses several baselines.

**Weaknesses:**

1. The claim that previous latent concatenation is impractical (L313-L316) lacks experimental support. To proof this, the authors should report the impact on GPU memory during training and inference, as well as the performance under an equivalent computational budget compared to the current method.

2. Since the proposed Motion Semantic Extractor (MSE) uses CLIP as its backbone, it may inherit CLIP's limitation in capturing fine-grained details. Consequently, the MSE risks failing to accurately model subtle motion information from the previous video chunk.

3. There're severe artifacts in the generated videos on the webpage. For example, the wheel in the moving car example moves unnaturally.

**Questions:**

Is multi-stage training necessary? Have you experimented with training all the modules end-to-end in a single stage?

---

> ### Author Response · Authors · 2025-11-24
>
> We appreciate your comments and hereby provide the following responses to thoroughly address your concerns:
>
> ### Q1: Experimental support for latent concatenation
>
> **R1**: As presented in the table below, we compare the computational overhead at inference and performance metrics of the two methods when conditioning on 8 previous frames and generating following 16 frames. Evidently, our decoupled approach exhibits substantial advantages in both performance and computational efficiency. Notably, when generating the same number of frames, it is impossible to enforce identical computational complexity between the two algorithms. **However, the fact that the latent concatenation method incurs significantly higher computational costs while achieving inferior performance—when compared to our approach—sufficiently demonstrates the effectiveness of our proposed method.**
> | Setting            | Inference Time | GPU Memory |FVD₃₂ₓ (↓) | VBench₃₂ₓ (↑) |
> | ------------------ | -------------- | ---------- |---------- | ------------- |
> | TaRoPE+Latent Cond | 4435ms         | 13GB |336.3      | 0.8277        |
> | TaRoPE+Ours        | 2536ms         | 8GB |319.9      | 0.8324        |
>
> ### Q2: CLIP's limitations
>
> **R2**: First and foremost, in our design, the primary objective of the Motion Semantic Extractor (MSE) is to extract motion information at the semantic level, making CLIP an appropriate choice for this module. **Furthermore, several existing works, such as Blip3-o, have successfully utilized CLIP features for image generation, which validates CLIP's inherent capability to process and preserve low-level information.** Meanwhile, the significant performance improvements demonstrated in the ablation studies reported in Table 2 and Figure 6 further corroborate the effectiveness of our MSE design.
>
> ### Q3: Visual artifacts
>
> **R3**: We acknowledge that our model has not achieved visual perfection. **Nevertheless, we kindly request that the reviewers not overlook the fact that our method has achieved substantial improvements in visual quality compared to prior approaches.** Additionally, due to constrained computational resources, our base model is only scaled to 1.3B parameters, which has constrained the upper bound of its visual performance—an aspect we have thoroughly discussed in the limitations section (L1050-L1060).
>
> ### Q4: Effectiveness of multi-stage training
>
> **R4**: Regarding the effectiveness of multi-stage training, detailed experimental evidence has been provided in Table 4 of the Supplementary Materials (Lines 1026-1036). Specifically, the third row (corresponding to Stage 3 only) represents end-to-end single-stage training, while the last row corresponds to the complete multi-stage training paradigm. **Clear comparison results indicate that the multi-stage training approach yields distinct performance advantages over the single-stage counterpart.**

---

### Official Review · Reviewer_SVtL · 2025-11-01

**Soundness:** 3
**Presentation:** 3
**Contribution:** 3
**Rating:** 4
**Confidence:** 3

**Summary:**

The paper proposes ArbInterp, a generative VFI framework that supports synthesizing intermediate frames at arbitrary continuous timestamps and arbitrary sequence lengths. The core idea is to replace discrete temporal indices with normalized timestamps in temporal dimensions via a timestamp-aware Rotary Position Embedding (RoPE). The authors also introduce an appearance–motion decoupled conditioning strategy, where a Motion Semantic Extractor (MSE) helps maintain motion coherence through cross-attention. Experiments show consistent gains over generative VFI baselines across VBench dimensions.

**Strengths:**

- The method fine-tunes Wan 2.1 (1.3B) with relatively modest compute (50k videos, 8 GPUs), suggesting decent data efficiency for the reported gains.
- Replacing temporal indices with normalized timestamps in RoPE is simple, training-efficient, and directly addresses the fixed-length limitation in generative VFI.
- The figures and supplementary materials are clear and helpful for understanding and reproducing the approach.

**Weaknesses:**

- Model design constraints. Wan 2.1 uses a causal VAE, so the receptive field of the last latent can depend on previous content. This implies the encoder needs access to the whole clip, which seems at odds with arbitrarily interpolating between two standalone images. The paper should clarify this limitation and the exact inference procedure for two-image interpolation without full context.
- Limited qualitative evaluation. To better assess generalization, please include:
a: Interpolation results given two arbitrary images (without video context).
b: Multiple interpolation samples (different random seeds) for the same start/end frames to assess diversity and stability.
- The fps in the presented interpolations results is very low, weakening the author's claim of arbitrary-length interpolation.
- MultiInterpBench appears to be largely composed of DAVIS data. If so, its novelty as a benchmark is limited. Please clarify the composition and splits beyond reuse of DAVIS.
- Given that VBench automatic metrics may not fully match human perception, it would help to include more challenging and diverse cases beyond standard DAVIS examples (e.g., large motion, occlusions, motion blur, non-rigid motion, out-of-domain content), along with a small-scale user study if feasible.

**Questions:**

Please check Weaknesses

---

> ### Author Response · Authors · 2025-11-24
>
> We sincerely appreciate your recognition and valuable comments. Hereby, we provide the following responses to thoroughly address your concerns:
>
> ### Q1: Problem of causal VAE
>
> **R1**: To address the concern about the causal VAE, as highlighted in Lines 214–215 and 838–841 of our manuscript, we emphasize that **to simplify the correspondence between each latent and its timestamp, we only perform spatial compression during tokenization**. Put simply, each frame in our model is processed independently without video context during both training and testing. While this may increase the overall computational overhead, when combined with our proposed Segment-by-Segment Interpolation inference strategy, the computational complexity is effectively reduced to an acceptable level, as detailed in Lines 301–304.
>
> ### Q2: Limited qualitative evaluation
>
> **R2**: Regarding the limited qualitative evaluation:
>
>
>
> * a: As addressed in Q1, all examples we present are generated without relying on video context.
>
> * b: To verify the generated diversity and stability, we provide a new example in the Supplementary Materials, as illustrated in Figure 12. The motion of the examples in the figure is not fixed. It can be observed that our model generates distinct intermediate frames under different random seeds, demonstrating the **diversity** of our model. Meanwhile, each generated video follows similar motion trajectories, which verifies the **stability** of the generation results.
>
> ### Q3: Low FPS
>
> **R3**: To further validate our method’s capability to handle continuous timestamps, we present 256× comparisons in Table 3 (L985-995). Given that our original test set lacked sufficient frames for this granularity, we re-collected 10 videos specifically for this test. The results were obtained by averaging the performance across each 32-frame sub-clip (as most metrics cannot process sequences of such extended length). Even though the timestamp granularity exceeds our training range, the sustained performance advantage we observe further solidifies the effectiveness of our method in handling continuous timestamps.
>
> ### Q4: Evaluation improvement
>
> **R4**:
> 1. Dataset composition: As stated in Lines 887–889,  MultiInterpBench encompasses not only DAVIS but also SNU-FILM and XTEST—datasets widely recognized in the frame interpolation community for featuring large motion, occlusions, motion blur, and non-rigid motion. Thus, our data exhibits sufficient difficulty and enjoys community-wide credibility. Beyond MultiInterpBench, we also provide streaming frame interpolation examples on the anonymous website, which should satisfy the requirement for sufficient motion complexity.
>
> 2. Novelty: We would like to emphasize that the contribution of MultiInterpBench does not lie in its data composition, but rather in its testing setting for multi-rate frame interpolation. Prior works typically focus on testing at a fixed interpolation rate, whereas MultiInterpBench is the first benchmark to systematically evaluate multi-rate frame interpolation. This innovation fills a critical gap in the field’s evaluation methodology.
>
> 3. User study: We conducted a user study involving 28 participants and 50 cases, where users were asked to select the best model in terms of motion quality, frame fidelity, and temporal consistency. As shown in the table below, our method demonstrates distinct advantages over prior methods across all these dimensions:
>
> | Method          | Motion Quality | Frame Fidelity |Temporal Consistency |
> | ------------------ | -------------- | ---------- |---------- |
> | LDMVFI | 6.7% | 8.8% | 10.5% |
> | TRF        | 2.7% | 0.6% | 1.0% |
> | GI        | 3.6% | 4.1% | 1.1% |
> | DynamiCrafter        | 12.9% | 17.3% | 7.9% |
> | Ours        | 74.1% | 69.2% | 79.5% |

---

### Meta-Review · Area_Chair_EMSw · 2026-01-12

**Summary:**

Scores were initially low (4,2,4,6), with reviewers citing concerns about the novelty of the proposed position embedding, the fairness of comparisons against baselines, visual artifacts, and specific architectural constraints. The authors provided a comprehensive rebuttal that included new benchmark comparisons, user studies, and additional ablation experiments. One reviewer explicitly confirmed that all their concerns were addressed. Another reviewer acknowledged that issues regarding task definition and evaluation fairness were resolved but maintained reservations about the method's core novelty. The remaining two reviewers did not respond to the rebuttal.

**Reviewer Concerns:**

Reviewers posed a number of concerns, many of which were addressed in the rebuttal, though some points regarding novelty and visual quality remain outstanding due to a lack of reviewer follow-up or persistent disagreement.

- Reviewer PUsK argued that "Video Frame Interpolation" was misleading and suggested "Keyframe Interpolation." The authors agreed to this change.
- Concerns about missing baselines and unfair comparisons were resolved by the authors adding these to the evaluation.
- Reviewer E5JG's request for comparison against methods like Super SloMo was addressed with new data showing the proposed method's superiority.
- Requests to verify appearance-motion decoupling and handling of non-equidistant timestamps were addressed with new experiments.
- Reviewer PUsK remains unconvinced about the novelty of the positional embedding strategy, viewing it as a limited adaptation of existing concepts.
- Reviewer PUsK requested evidence of the method's utility in other generation tasks to bolster novelty claims. The authors promised results, but this remains an open point for the reviewer.
- Reviewer wVzm's strong concerns about artifacts (e.g., unnatural wheel movement) and CLIP's inability to capture fine details were rebutted by the authors but not confirmed as resolved by the reviewer.

**Reviewer Scores:**

Reviewer scores were initially low (4, 2, 4, 6) . Reviewer E5JG maintained their positive score (6) after the rebuttal. Although Reviewer wVzm did not engage with the rebuttal, the successful resolution of concerns for the most active reviewers suggests the final consensus leans marginally positive, likely settling around (4, 2, 5, 6) or (5, 2, 5, 6).

---

### Decision · Program_Chairs · 2026-01-26

Accept (Poster)